# Detecting Errors and Estimating Accuracy on Unlabeled Data with Self-training Ensembles

**Jiefeng Chen** *
Department of Computer Science
University of Wisconsin-Madison
Madison, WI 53706
jiefeng@cs.wisc.edu

**Frederick Liu**
Google
Seattle, WA 98103
frederickliu@google.com

**Besim Avci**
Google
Seattle, WA 98103
besim@google.com

**Xi Wu**
Google
Madison, WI 53703
wuxi@google.com

**Yingyu Liang**
Department of Computer Science
University of Wisconsin-Madison
Madison, WI 53706
yliang@cs.wisc.edu

**Somesh Jha**
Department of Computer Science
University of Wisconsin-Madison
Madison, WI 53706
jha@cs.wisc.edu

## Abstract

When a deep learning model is deployed in the wild, it can encounter test data drawn from distributions different from the training data distribution and suffer drop in performance. For safe deployment, it is essential to estimate the accuracy of the pre-trained model on the test data. However, the labels for the test inputs are usually not immediately available in practice, and obtaining them can be expensive. This observation leads to two challenging tasks: (1) *unsupervised accuracy estimation*, which aims to estimate the accuracy of a pre-trained classifier on a set of unlabeled test inputs; (2) *error detection*, which aims to identify mis-classified test inputs. In this paper, we propose a principled and practically effective framework that simultaneously addresses the two tasks. The proposed framework iteratively learns an ensemble of models to identify mis-classified data points and performs self-training to improve the ensemble with the identified points. Theoretical analysis demonstrates that our framework enjoys provable guarantees for both accuracy estimation and error detection under mild conditions readily satisfied by practical deep learning models. Along with the framework, we proposed and experimented with two instantiations and achieved state-of-the-art results on 59 tasks. For example, on iWildCam, one instantiation reduces the estimation error for unsupervised accuracy estimation by at least 70% and improves the F1 score for error detection by at least 4.7% compared to existing methods.

---
*Part of the work done while interning at Google.
Our code is available at: https://github.com/jfc43/self-training-ensembles.

35th Conference on Neural Information Processing Systems (NeurIPS 2021).

# 1 Introduction

Data distribution in the real world may be wildly different from the training dataset for various reasons such as covariate shift due to domain divergence, corruption of images due to weather conditions, or out-of-distribution test inputs. When facing these issues, a deployed deep learning model can have unexpected performance drop on the test data. This performance degradation can be mitigated by test data annotation, which may be costly. Hence, estimating the accuracy of a pre-trained model on the unlabeled test data provides an alternative to avoid the cost when it is not necessary. Furthermore, it is beneficial to estimate the correctness of the predictions on individual points. This leads to an even more challenging task of error detection, which aims to identify points in the unlabeled test set that are mis-classified by the pre-trained model. Such a finer-grained estimation can facilitate a further improvement of the pre-trained model (e.g., manually label those mis-classified data points and retrain the model on them).

While there have been previous attempts to address accuracy estimation or the broader problem of error detection, their successes usually rely on some conditions or assumptions that may not hold in practice. For example, a natural approach is to use confidence based metrics to measure the performance of the pre-trained model, e.g., as in [9]. If the model is well-calibrated on the test data, then the average confidence approximates its accuracy. However, it has been observed that many machine learning systems, in particular modern neural networks, are poorly calibrated, especially on test data with distribution shift [14, 30]. Another method is to learn a regression function that takes statistics about the model and the test data as input, and predicts the performance on the test data [9, 35]. This requires training on labeled data from various data distributions, which is very expensive or even impractical. Furthermore, the performance predictor trained on the labeled data may not generalize to unknown data distributions. Recent work by [4] proposes to learn a "check" model using domain-invariant representation and use it as a proxy for the unknown true test labels to estimate the performance via error detection. It relies on the success of the domain-invariant representation methods to obtain a highly accurate check model on the test data. Hence, the check model performance suffers when domain-invariant representation is not accurate in circumstances such as test data having outlier feature vectors or different class probabilities than the training data.

In this paper, we propose a principled and practically effective framework for the challenging tasks of accuracy estimation and error detection (Section 4). The framework makes a novel use of the self-training technique on ensembles for these tasks. It first learns an ensemble of models to identify some mis-classified points. Then it assigns pseudo-labels to these points and uses self-training with these pseudo-labeled data to identify more mis-classified points. We also provide provable guarantees for the framework (Section 5). Our analysis shows that it provably outputs an accurate estimate of the accuracy as well as the mis-classified points, under mild practical conditions: the ensemble make small errors on the test inputs correctly classified by the model $f$, mostly disagree with $f$ on the current identified mis-classified inputs, and are correct or have diverse predictions on the remaining inputs. These conditions can be readily satisfied by deep learning model ensembles. Furthermore, they have no explicit assumptions on the test distribution and thus the framework can be instantiated by incorporating properly designed ensemble methods for different settings (Section 6). Experimental results on 59 tasks over five dataset categories including image classification and sentiment classification datasets show that our method achieves state-of-the-art on both accuracy estimation and error detection (Section 7). The experiments also provide positive support for our analysis, verifying the conditions and implications.

# 2 Related Work

**Confidence Estimation and Error Detection.** Recently, estimating model confidence has been an important area of research because of the perceived relationship between model uncertainty and trusting its predictions [28]. However, modern neural networks have been observed to be poorly calibrated (e.g. they may make wrong predictions with very high confidence) [14], especially on distributions with dataset shift [30]. Many approaches have been proposed to address this issue, such as *Temperature Scaling* [14], *Monte-Carlo Dropout* [11] and *Deep Ensemble* [25], but the challenge still remains. A similar challenge appears in error detection, where the goal is to identify erroneous predictions given a test set. Combining these two problems, some early work uses confidence estimates to detect incorrect predictions. For example, [17] proposed to use maximum

softmax probability to detect misclassified examples. [5] proposed *True Class Probability* for failure prediction. [21] proposed to use *Trust Score* to estimate the confidence in model predictions. These methods require a robust estimate of confidence, and an empirically chosen threshold that is mostly problem- and model-dependent, to identify error data points. Recently, [4] proposed to use a check model to predict mis-classification.

**Unsupervised Accuracy Estimation.** The problem of unsupervised accuracy (or risk) estimation has received relatively scant attention from the research community. [8] offered a solution with certain assumptions on the marginal output distribution $p(y)$. [32] proposed to estimate the accuracies of the approximations to some target Boolean functions based on the agreement rates method. A follow-up work by [31] also considered the "multiple approximations" problem setting where the target classes may be tied together through logical constraints, and proposed an efficient method to estimate the accuracy of classifiers using only unlabeled data. [20] proposed a spectral-based approach to estimate the accuracies of multiple classifiers, mainly in the binary case and with some assumptions on the classifiers. [36] proposed a method to estimate the model's error on distributions different from the training distribution by assuming a conditional independence structure of the data. These problem settings are different from ours and the constraints may not be satisfied by the data and model we consider. Recently, [9] studied three families of methods ($\mathcal{H}$-divergence, reverse classification accuracy and confidence measures) and demonstrated how they could be used to predict the performance drop. [35] also proposed to learn a performance predictor on the statistics of model outputs with an assumption that they know the typical cases of dataset shift in advance. Similarly, [7] proposed to train regression models on the feature statistics of the datasets sampled from a meta-dataset to predict model performance and [6] proposed to utilize linear regression to estimate classifier performance from the accuracy of rotation prediction. [13] proposed to use difference of confidences to estimate classifier performance. [3] proposed MANDOLINE that utilizes user-specified slicing functions to improve the importance of weighting to make the accuracy estimation more accurate. With the most similar setup to ours, [4] used a set of domain-invariant predictors as a proxy for the unknown target labels to estimate a given model's performance under distribution shift. In a concurrent work, [22] empirically showed that the test accuracy of deep networks can be estimated by measuring disagreement rate between a pair of models independently trained via Stochastic Gradient Descent (SGD) and theoretically related this phenomenon to the well-calibrated nature of ensembles of SGD-trained models.

## 3 Problem Statement

Consider the classification problem with sample space $\mathcal{X}$ and label space $\mathcal{Y}$. Let $D$ be a set of labeled data from the training distribution $P_{X,Y}$. Let $U = \{(x, y_x)\}$ be a set of labeled data from the test distribution $Q_{X,Y}$, where $y_x$ is the label for $x$. Let $U_X = \{x : (x, y_x) \in U\}$ be the set of input feature vectors in $U$. Define the accuracy of a model $f$ on $U$ as: $\mathrm{acc}(f, U) := \frac{1}{|U|} \sum_{(x,y_x) \in U} \mathbb{I}[f(x) = y_x]$, where $|U|$ is the cardinality of the set $U$. When clear from the context, we will omit $U$ and simply write $\mathrm{acc}(f)$. Given a model $f : \mathcal{X} \to \mathcal{Y}$ trained on $D$, together with $D$ and $U_X$, the goal of *unsupervised accuracy estimation* is to get an estimate $\mathrm{a\hat{c}c}$ so that the absolute estimation error $|\mathrm{a\hat{c}c} - \mathrm{acc}(f)|$ is small. We assume access to the training data $D$ to help in estimating $\mathrm{acc}(f)$.

Though $\mathrm{acc}(f)$ is usually sufficient to provide an estimate of whether the model could perform well on $U$, it would be beneficial if the algorithm could also provide an estimate of the correctness of individual points so that we can know where to improve the model $f$. Thus, we also consider a more challenging task of identifying the mis-classified points in $U$: $W_X := \{x : (x, y_x) \in U, f(x) \neq y_x\}$. Given $f$, $D$ and $U_X$, the goal of *error detection* is to identify a subset $R_X \subseteq U_X$, such that $|W_X \triangle R_X|$ is small, where $W_X \triangle R_X := (W_X \setminus R_X) \cup (R_X \setminus W_X)$.

## 4 Algorithmic Framework via Self-Training Ensembles

**Intuition.** We consider the approach of learning a "check model" $h$ and using the disagreement between $h$ and the pre-trained model $f$ for our tasks. The fundamental idea is to identify a point $x$ as mis-classified if $h$ disagrees with $f$ on $x$. To derive our method, we note a simple fact: the disagreement approach succeeds if (1) $h$ agrees with $f$ on points where $f$ is correct and (2) $h$ disagrees with $f$ on points where $f$ is incorrect. Our first key observation is that usually (1) can

**Framework 1** Error Detection and Unsupervised Accuracy Estimation via Self-Training Ensembles

---

**Input:** A training dataset $D$, an unlabeled test dataset $U_X$, a pre-trained model $f$, an ensemble learning method $\mathcal{T}$, and a hyper-parameter $\tau$
1: Initialize $R = \emptyset$
2: **for** $t = 1, 2, \cdots, T$ **do**
3:    Use $\mathcal{T}$ to generate an ensemble $\mathcal{T}(D, U_X, R)$.
4:    Identify those points on which the ensemble and $f$ disagree as mis-classified points:

$$R_X := \{x \in U_X : \Pr_{h \sim \mathcal{T}}\{h(x) = f(x)\} < \tau\}.$$

5:    For each $x \in R_X$, assign a pseudo-label $\tilde{y}_x \neq f(x)$ (e.g., using the majority vote of the ensemble or a random label). And set $R = \{(x, \tilde{y}_x) : x \in R_X\}$.
6: **end for**
**Output:** The estimated mis-classified points $R_X$, and the estimated accuracy $\frac{|U_X \setminus R_X|}{|U_X|}$.

---

be satisfied approximately in practice. Intuitively, this is because $h$ and $f$ use the same training data, and $h$ can be trained to be correct on the subset of the instance space where $f$ is correct. However, (2) may not be easily satisfied (e.g., $h$ can make similar mistakes as $f$), which leads to an overestimation of the accuracy. We thus focus on improving the disagreement on mis-classified points.

To disagree with $f$ on a mis-classified test input $x$, we have two ways: (1) make the check model $h$ correct on $x$; (2) diverse ensembles. The first way may be achievable when the training data contains information for prediction on $x$. A prototypical example is when the test inputs are corruption of clean data from the training distribution (e.g., the training data are images in sunny days while the test inputs are ones in rainy days), and techniques like unsupervised domain adaptation can be used to improve the prediction on such test inputs. However, correct predictions on $x$ may not be feasible in many interesting scenarios due to insufficient information (e.g., the test image in the open world can contain an object that is never seen in the training data ). Fortunately, it has been shown that for such inputs, one can obtain an ensemble of models with diverse predictions (e.g.,[25]). This then gives the second way to achieve disagreement: using diverse ensembles. Therefore, our method will learn an *ensemble* of models (instead of one check model) and identify a point $x$ as mis-classified if the ensemble disagree with $f$ on $x$ (i.e., a large fraction of the models in the ensemble disagree with $f$ on their predictions on $x$). A mis-classified test input, $x$, will be successfully identified, if a majority of the ensemble models predict correctly, or they have large diversity on $x$.

However, the ensemble may only be able to identify a subset of the mis-classified points. Therefore, we propose to iteratively identify more and more mis-classified points by *self-training*. For each mis-classified data point $x$ identified by the ensemble, we assign it a pseudo-label that is different from $f(x)$ (e.g. use the majority vote of the ensemble or a random label as the pseudo-label). Then we can train (with regularization) a new ensemble to encourage their disagreement with $f$ on the pseudo-labeled data $R$ (e.g., use a supervised loss on $R$ with a small weight as the regularization). Note that the pseudo-labels may not all be correct, and we do not need the new ensemble to exactly fit the pseudo-labels. We only need the new ensemble to mostly disagree with $f$ on $R$ so that they still identify $R$ as mis-classified points. Furthermore, self-training can help the ensemble identify more mis-classified points. When some pseudo-labels are correct, we observe that these additional data can help the new ensemble become more accurate and thus help with identifying more mis-classified points. We also observe that the new ensemble will be less diverse on the pseudo-labeled data and thus have diversity on the remaining test inputs. We report empirical results in Appendix C.4 to support these observations.

**Our framework.** We assume access to an ensemble learning method $\mathcal{T}$ (specific instantiations will be given later) that takes as input a training set $D$, an unlabeled test set $U_X$, and a pseudo-labeled set $R$ (and potentially some other parameters) and outputs an ensemble, which is a distribution over functions $h : \mathcal{X} \to \mathcal{Y}$.[2] We denote the generated ensemble as $\mathcal{T}(D, U_X, R)$ and write $h \sim \mathcal{T}(D, U_X, R)$ for sampling $h$ from the ensemble, or simply $h \sim \mathcal{T}$ when clear from context.

---

[2]This includes one single $h$ as a special case. It also includes the case of $h$ with probabilistic outputs, i.e., $h(x) = [h^1(x), \ldots, h^K(x)]$ for $\mathcal{Y} = \{1, 2, \ldots, K\}$, where $h^i(x)$ is the predicted probability of $x$ from class $i$. One can think of such an $h$ as an ensemble $\mathcal{T}$ where $\Pr_{g \sim \mathcal{T}}[g(x) = i] = h^i(x)$.

We are now ready to describe our method (Framework 1). Our framework begins with an empty set of pseudo-labeled data $R$, and executes in $T$ iterations. In each iteration, it generates an ensemble $\mathcal{T}(D, U_X, R)$ and then uses the ensemble to construct a new pseudo-labeled dataset $R$ for the next iteration: let $R_X$ be those points $x \in U_X$ where the agreement rate between the ensemble and $f$ is below a threshold, assign a pseudo-label $\tilde{y}_x$ different from $f(x)$, and let $R$ be the set of these pseudo-labeled data. Finally, it outputs $R_X$ as the mis-classified points and outputs the fraction of points outside $R_X$ as the accuracy.

Some existing work also use models' disagreement to estimate the error on the unlabeled data in different ways. For example, [32] consider a set of pre-trained models and aim to estimate the error for each model; [4] learn a single check model and compare it with $f$. On the other hand, our novelty is using *ensembles* to identify a set of mis-classified points and further using *self-training* for learning the ensembles. Note that our self-training ensembles is different from standard self-training and ensemble: standard self-training+ensemble aims to get accurate predictions while ours is to increase disagreement on mis-classified points (refer to Appendix A.2 for a detailed discussion).

# 5   Theoretical Analysis

As described in the intuition above, our framework succeeds if in each iteration, the ensemble satisfy the following conditions: **(A)** correct on $U_X \setminus W_X$, **(B)** mostly disagree with $f$ on $R_X$, and **(C)** either correct or diverse on $W_X \setminus R_X$. This section first introduces notions formalizing these conditions and then provides provable guarantees under these conditions. Here we focus on intuitions and provide the proofs and more discussion in Appendix B.

**Notations.** Recall that the ensemble method $\mathcal{T}$ outputs a distribution over models, and we write $h \sim \mathcal{T}$ for sampling a model $h$ from this distribution. Let $\mathbb{E}_h$ denote the expectation over this output distribution, and $\mathbb{E}_x$ denote the average over $x \in U_X$ or $(x, y_x) \in U$.

**Formalizing Condition (A).** A key observation we make in practice is that usually on points where $f$ is correct, the ensemble models are also correct. Intuitively, this is because the only labeled information for learning the ensemble is $D$, while $D$ is also used for learning $f$. For illustration, suppose $f$ and the ensemble have zero training errors on $D$. When they are from hypothesis classes of bounded VC-dimensions and the training set is large enough compared to the VC-dimensions, standard error bounds show that the ensemble will have small errors on the subset of the feature space $\{x \in \mathcal{X} : f(x) = y_x\}$, i.e., it will have small errors on points where $f$ is correct. To formalize this observation, we introduce the following notion.

**Definition 1** (Error on Correct Points). *Let $\nu$ denote the average probability of $h \sim \mathcal{T}$ making error on test inputs where the model $f$ is correct:* $\nu := \Pr_{(x,y_x) \sim U, h \sim \mathcal{T}} [h(x) \neq y_x \mid f(x) = y_x]$.

**Formalizing Condition (B).** We assume the new ensemble trained with regularization on $R$ will mostly disagree with $f$ on $R_X$. We define the following notion:

**Definition 2** (Agreement on Pseudo-Labeled Data). *Let $\gamma$ denote the average probability of agreement between $h \sim \mathcal{T}$ and $f$ on $R_X$:* $\gamma := \Pr_{x \sim R_X, h \sim \mathcal{T}} \{h(x) = f(x)\}$.

**Formalizing Condition (C).** Let $G_X$ denote the good points in $W_X \setminus R_X$ on which the ensemble will have correct predictions with high confidence, $B_X$ the remaining bad points. Formally, we define: $G_X := \{x \in W_X \setminus R_X : \Pr_{h \sim \mathcal{T}}\{h(x) = y_x\} \geq 1 - \nu\}$ and $B_X := W_X \setminus (R_X \cup G_X)$.

(Here we choose the confidence level $1 - \nu$ for convenience, where $\nu$ is the error on correct points. Other sufficiently high confidence levels can also be used with slight change to the analysis.) We would like the ensemble to have large diversity on $B_X$.

**Definition 3** (Diversity of Ensemble). *Let $\sigma^2$ denote the average probability of disagreement between two ensemble models on $B_X$:* $\sigma^2 := \mathbb{E}_x[\sigma_x^2 | x \in B_X]$, where $\sigma_x^2 := \Pr_{h_1, h_2 \sim \mathcal{T}}[h_1(x) \neq h_2(x)]$.

**Provable Guarantees.** Based on the above notions we obtain our guarantees:

**Theorem 1.** *Assume in each iteration of the framework, $\tau = \sqrt{1 - \eta}$ for some $\eta \in (0, 3B_\eta/4)$ where $B_\eta := \min\{\sigma^2, 1 - \nu^2\}$. Let $\sigma_L^2 > 0$ be a lower bound on the diversity $\sigma^2$, $\tilde{\gamma} > 0$ be an upper bound on $\gamma$, and $\tilde{\nu}$ be an upper bound on $\nu$ over all iterations. Then for any $\delta \in (0, \sigma_L^2/4)$, after at*

**Algorithm 1** Error Detection and Unsupervised Accuracy Estimation via Self-Training Ensembles

**Input:** $D, U_X, f$, ensemble method $\mathcal{T}$, parameters $T, N$
 1: Initialize $R = \emptyset$
 2: **for** $t = 1, 2, \cdots, T$ **do**
 3:     Set $\{h_i\}_{i=1}^N = \mathcal{T}(D, U_X, R, N, \text{other parameters})$
 4:     Let $\tilde{y}_x$ be the majority vote of $\{h_i\}_{i=1}^N$: $\tilde{y}_x := \arg\max_{j \in \mathcal{Y}} \frac{1}{N} \sum_{i=1}^N \mathbb{I}[h_i(x) = j]$.
 5:     Set $R = \{(x, \tilde{y}_x) : x \in U_X, \tilde{y}_x \neq f(x)\}$.
 6: **end for**
**Output:** Estimated mis-classified points $R_X = \{x : (x, y) \in R\}$, estimated accuracy $\frac{|U_X \setminus R_X|}{|U_X|}$.

---

*most $\lceil 1/\delta \rceil$ iterations, we can get $\frac{|U_X \setminus R_X|}{|U_X|}$ approximates the accuracy* $\mathrm{acc}(f)$ *and $R_X$ approximates the mis-classified points $W_X$ as follows:*

$$\left| \mathrm{acc}(f) - \frac{|U_X \setminus R_X|}{|U_X|} \right| \leq \max\{\frac{\tilde{\nu}}{1-\tau}(1 - e_f), \epsilon e_f\}, \; where \; \epsilon := \frac{\frac{\tilde{\gamma}}{\tau}\left(1 + \frac{\tilde{\nu}}{1-\tau}\frac{1-e_f}{e_f}\right)}{\frac{\sigma_L^2}{4} - \delta + \frac{\tilde{\gamma}}{\tau}}, \quad (1)$$

$$|W_X \triangle R_X| \leq \frac{\tilde{\nu}}{1-\tau}|U_X \setminus W_X| + \epsilon|W_X|. \quad (2)$$

*Proof Sketch.* Let's first consider one iteration, assuming small $\nu, \gamma$ and large $\sigma^2$. Intuitively, on the correct points $U_X \setminus W_X$, the ensemble have a small error $\nu$ and thus disagree with $f$ on only a few such points. On the old $R_X$, the ensemble mostly disagree with $f(x)$; similarly on $G_X$. On the remaining points $B_X$, we show that if the ensemble have large diversity, then their predictions must have large variances on a significant subset of points and thus have large disagreement with $f$, facilitating the detection. Overall, our framework can construct a new pseudo-labeled set $R'_X$ that contains mostly mis-classified points and is also larger than the old $R_X$ (Lemma 1 in Appendix B). Therefore, each iteration can make some progress by identifying more mis-classified points than before, and enough iterations achieve the guarantees. $\quad \square$

The theorem provides guarantees for general values of $\nu, \gamma$ and $\sigma^2$. To get some intuition, note that typically $e_f < \tau$ and suppose we set $\tau = 3/4$ and $\delta = \tilde{\gamma}/\tau$, then the accuracy is estimated up to error $\max\{\tilde{\nu}, \frac{16\tilde{\gamma}}{3\sigma_L^2}(e_f + \tilde{\nu})\}$. When $\tilde{\nu}, \tilde{\gamma}$ are small and $\sigma_L^2$ is large, the error is small. Therefore, under mild conditions, our framework can give a provable estimation of the accuracy and the mis-classified points up to small errors.

The theorem formalizes that the framework can succeed under mild conditions **(A)(B)(C)** on the ensembles, without explicit conditions on the data distributions and the pre-trained model (note that the conditions on them are implicitly captured in the mild conditions on the ensemble). The framework is thus flexible, and different ensemble methods satisfying these conditions can be incorporated to get concrete instantiations applicable to various settings. Indeed, the crux of our framework is then to design ensemble methods meeting the conditions. This turns out to be not difficult, in particular for deep learning pre-trained models and deep learning ensemble models. Even an ensemble of deep networks simply trained from random initialization works well. Two concrete instantiations are presented in Section 6 below and evaluated in Section 7.

## 6 Instantiations of the Framework

Based on our framework, we propose Algorithm 1 for accuracy estimation and error detection. It executes in $T$ iterations. In each iteration, it trains an ensemble of models $\{h_i\}_{i=1}^N$ and then uses the ensemble to construct the pseudo-labeled data. Finally, it outputs $R_X$ as the set of mis-classified points and outputs the fraction of points outside $R_X$ as the accuracy. The algorithm uses an intuitive heuristic to implement the threshold on the agreement rate: it sets $R_X$ as the points $x \in U_X$ with $f(x)$ different from the majority vote of the ensemble models. Empirically, we observe that this leads to similar $R_X$ and thus similar results as explicit thresholding, but is much more convenient (the empirical results to support this observation are reported in Appendix C.6).

---

**Algorithm 2** $\mathcal{T}_{\text{RI}}$: Ensemble via Random Initialization

---

**Input:** $D$, $U_X$, $R$, $N$, parameter $\gamma$
  1: Pre-train $\{h_i'\}_{i=1}^N$ on $D$ from different random initialization
  2: **for** $i = 1, 2, \ldots, N$ **do**
  3:     Learn $h_i$ by fine-tuning $h_i'$ for one epoch by:

$$\underset{h}{\text{minimize}} \quad \mathbb{E}_{(x,y)\in D}[\ell(h(x), y)] + \gamma \cdot \mathbb{E}_{(x,y)\in R}[\ell(h(x), y)] \tag{3}$$

  4: **end for**
**Output:** The ensemble of models $\{h_i\}_{i=1}^N$

---

---

**Algorithm 3** $\mathcal{T}_{\text{RM}}$: Ensemble via Representation Matching

---

**Input:** $D$, $U_X$, $R$, $N$, initial pre-trained model $h_0$, parameters $\alpha, \gamma$
    *// $h_0(x) = c(\phi(x))$ is a composition of a prediction function $c$ and a representation function $\phi$*
  1: Fine-tune $h_0$ for $N$ epochs using the objective:

$$\underset{h}{\text{minimize}} \quad \mathbb{E}_{(x,y)\in D}[\ell(h(x), y)] + \gamma \cdot \mathbb{E}_{(x,y)\in R}[\ell(h(x), y)] + \alpha \cdot d(p_D^\phi, p_{U_X}^\phi) \tag{4}$$

    where $d(p_D^\phi, p_{U_X}^\phi)$ is the distance between the distribution of $\phi(x)$ on $D$ and that on $U_X$.
  2: Use the $N$ checkpoint models at the end of each training epoch as the model ensemble $\{h_i\}_{i=1}^N$.
**Output:** The ensemble of models $\{h_i\}_{i=1}^N$

---

While different ensemble methods $\mathcal{T}$ can be used in Algorithm 1, here we propose two concrete instantiations $\mathcal{T}_{\text{RI}}$ and $\mathcal{T}_{\text{RM}}$ based on the success conditions of our framework.

**Ensemble Method $\mathcal{T}_{\text{RI}}$.** Algorithm 2 describes a natural method that trains the models from different random initialization. It first trains $h_i'$ on $D$ (e.g., using the same training algorithm as for $f$), to ensure that ensemble has small error on points where $f$ is correct. It then fine-tunes $h_i'$ on $D$ and $R$ for one epoch to get $h_i$ that mostly disagrees with $f$ on $R_X$. Finally, deep models trained from different random initialization have been shown to be diverse on outlier data points [25, 10]. In summary, the ensemble constructed can satisfy our three conditions.

**Ensemble Method $\mathcal{T}_{\text{RM}}$.** Algorithm 3 describes another method designed with the representation matching technique for domain adaptation, which can potentially improve the accuracy of the ensemble on some test inputs related to the training data and thus satisfy our success condition **(C)** better. It requires the model architecture to be $c(\phi(x))$, i.e., a composition of a prediction function $c$ and a representation function $\phi$. Beginning from a pre-trained model $h_0$, it fine-tunes $h_0$ for $N$ epochs by minimizing the loss on $D$ and $R$ *plus a representation matching loss* $\alpha \cdot d(p_D^\phi, p_{U_X}^\phi)$, and outputs the $N$ checkpoint models at the end of each training epoch as the ensemble. A key component of $\mathcal{T}_{\text{RM}}$ is representation matching, which aims to learn a function $\phi$ that minimizes $d(p_D^\phi, p_{U_X}^\phi)$. It has been shown that representation matching can improve the accuracy on the test data from the target domain. Also, we have observed that the checkpoint models can have diversity on the misclassified data points empirically. Thus, the ensemble constructed can satisfy our conditions. For our experiments, we use the representation matching loss from the classic DANN [12].

## 7 Experiments

We perform experiments for unsupervised accuracy estimation and error detection tasks on 59 pairs of training-test datasets from five dataset categories, including image classification and sentiment classification datasets. In summary, our findings are: **(1)** Our method achieves state-of-the-art results on both accuracy estimation and error detection tasks than the existing methods. **(2)** Both ensemble and self-training techniques have positive effects on the tasks and it is easy to pick suitable hyper-parameters for our algorithms. **(3)** Empirical results show that the conditions made in our analysis hold approximately.

### 7.1 Setup

We briefly describe the experiment setup here. The detailed setup can be found in Appendix C.1.

**Dataset.** The task needs a pair of training-test datasets $D$ and $U_X$. We use five dataset categories, each containing multiple training-test dataset pairs. Specifically, we use the following dataset categories: Digits (including MNIST [26], MNIST-M [12], SVHN [29], USPS [19]), Office-31 [33], CIFAR10-C [24], iWildCam [1] and Amazon Review [2]. Digits has 12 dataset pairs, Office-31 has 6, CIFAR10-C has 19, iWildCam has 10 and Amazon Review has 12.

**Evaluation Metrics.** We use absolute estimation error for accuracy estimation and use F1 score for error detection.

**Our Models.** On each dataset category, we design a neural network architecture for the DANN training algorithm [12], which is named DANN-arch. It contains an encoder, a predictor branch and a discriminator branch. For $\mathcal{T}_{RM}$, we use the DANN-arch for the ensemble $\{h_i\}$ and pre-train the initial model $h_0$ on $D$ and $U_X$ using DANN algorithm. For $\{h_i\}$ in $\mathcal{T}_{RI}$, we also use the DANN-arch. The model $f$ is pre-trained on $D$ and we mainly consider two kinds of architectures for it: one is the DANN-arch (i.e., the model $f$ shares the same architecture as the check model $h$); the other is a typical deep neural network (DNN).

**Hyper-parameters.** The hyper-parameters $T$, $N$ and $\gamma$ can be easily selected since a broad range of values can lead to good results. Based on our observation, larger $T$ and $N$ can lead to better results. In our experiments, we set $T = 5$ and $N = 5$ by considering the computational cost (on Amazon Review, we set $N = 20$). We set $\gamma = 0.1$ and set $\alpha$ following the domain adaptation methods.

**Baselines.** For accuracy estimation, we consider Proxy Risk [4], Average Confidence (Avg Conf) [9], and Ensemble Average Confidence (Ens Avg Conf). For error detection, we consider Proxy Risk, Maximum Softmax Probability (MSP) [17], and Trust Score [21]. Although Proxy Risk and our method share some similar ideas such as the use of check models, disagreement and representation matching, they have some major differences in the key ideas, training objectives, and the implementation of training objectives (refer to Appendix A.1 for a detailed discussion).

### 7.2 Results

**Performance for Accuracy Estimation.** The results in Table 1 show that our method (with $\mathcal{T}_{RM}$) achieves significantly better results across various dataset categories compared to existing methods. Specifically, on Digits and CIFAR10-C, it outperforms current state-of-the-art method Proxy Risk significantly (e.g. reduce the error by $> 40\%$). On Office-31 and Amazon Review, it also outperforms the others and has a large advantage on the pre-trained models using DANN-arch.

We would like to emphasize the results on the challenging dataset iWildCam. Our method (with either $\mathcal{T}_{RI}$ or $\mathcal{T}_{RM}$) outperforms the other methods significantly (e.g., reduce the error by $> 70\%$), and the instantiation with $\mathcal{T}_{RI}$ gets the best results. Note that on iWildCam, the label distribution of the test data is imbalanced and different from that of the training data. In such a case, the representation matching technique will fail since the representations of the two domains may be misaligned. Thus, the performance of Proxy Risk becomes worse, as it relies on representation matching. Our method with $\mathcal{T}_{RM}$ also uses DANN in the ensemble method, but performs significantly better than Proxy Risk, since the diversity helps satisfy condition **(C)** though the representation matching fails to improve accuracy there. This shows that our ensemble and self-training techniques could alleviate the drawbacks of representation matching in such cases. Furthermore, our method with $\mathcal{T}_{RI}$ achieves even better results, which demonstrates the flexibility and effectiveness of our framework.

The results for each dataset pair from Digits and CIFAR10-C are plotted in Figure 1. The plots show that proxy risk tends to underestimate the accuracy. This is because proxy risk maximizes disagreement which can overly suppress the accuracy. While the average confidence methods tend to overestimate the accuracy, because the model $f$ tends to be overconfident on the data with dataset shift, even when this issue gets rectified in the ensemble average confidence method. In comparison, our method with $\mathcal{T}_{RM}$ exhibits a clear advantage. The full results for all dataset categories and pre-trained models in Appendix C.2 show a similar trend.

**Performance for Error Detection.** Table 1 shows that our method with $\mathcal{T}_{RM}$ outperforms existing methods on all dataset categories. Specifically, our method improves the F1 score by at least 4.4%

| Task | | Accuracy Estimation | | | Error Detection | | |
|---|---|---|---|---|---|---|---|
| Metric | | Absolute Estimation Error ↓ | | | F1 score ↑ | | |
| Dataset | Method | Model $f$ | | Method | Model $f$ | | |
| | | Typical DNN | DANN-arch | | Typical DNN | DANN-arch | |
| Digits | Avg Conf | 0.404±0.180 | 0.350±0.230 | MSP | 0.467±0.195 | 0.485±0.209 | |
| | Ens Avg Conf | 0.337±0.229 | 0.246±0.230 | Trust Score | 0.496±0.195 | 0.484±0.187 | |
| | Proxy Risk | 0.085±0.142 | 0.095±0.181 | Proxy Risk | 0.844±0.118 | 0.796±0.155 | |
| | Ours (RI) | 0.164±0.218 | 0.087±0.077 | Ours (RI) | 0.698±0.235 | 0.701±0.126 | |
| | Ours (RM) | **0.023**±0.020 | **0.024**±0.022 | Ours (RM) | **0.881**±0.084 | **0.841**±0.112 | |
| Office-31 | Avg Conf | 0.054±0.044 | 0.259±0.134 | MSP | 0.281±0.266 | 0.584±0.128 | |
| | Ens Avg Conf | 0.080±0.041 | 0.281±0.136 | Trust Score | 0.401±0.240 | 0.559±0.143 | |
| | Proxy Risk | 0.033±0.012 | 0.042±0.034 | Proxy Risk | 0.605±0.177 | 0.629±0.140 | |
| | Ours (RI) | 0.051±0.038 | 0.044±0.031 | Ours (RI) | 0.715±0.124 | 0.770±0.027 | |
| | Ours (RM) | **0.029**±0.021 | **0.018**±0.023 | Ours (RM) | **0.767**±0.052 | **0.790**±0.087 | |
| CIFAR10-C | Avg Conf | 0.353±0.175 | 0.369±0.176 | MSP | 0.505±0.043 | 0.550±0.043 | |
| | Ens Avg Conf | 0.237±0.144 | 0.237±0.133 | Trust Score | 0.494±0.045 | 0.568±0.060 | |
| | Proxy Risk | 0.053±0.070 | 0.052±0.070 | Proxy Risk | 0.850±0.107 | 0.843±0.101 | |
| | Ours (RI) | 0.149±0.089 | 0.197±0.115 | Ours (RI) | 0.654±0.064 | 0.568±0.063 | |
| | Ours (RM) | **0.022**±0.009 | **0.029**±0.012 | Ours (RM) | **0.872**±0.083 | **0.860**±0.091 | |
| iWildCam | Avg Conf | 0.388±0.045 | 0.395±0.043 | MSP | 0.692±0.006 | 0.741±0.009 | |
| | Ens Avg Conf | 0.177±0.025 | 0.158±0.020 | Trust Score | 0.717±0.009 | 0.737±0.010 | |
| | Proxy Risk | 0.119±0.043 | 0.094±0.036 | Proxy Risk | 0.755±0.038 | 0.773±0.039 | |
| | Ours (RI) | **0.015**±0.008 | **0.007**±0.004 | Ours (RI) | 0.792±0.013 | 0.806±0.012 | |
| | Ours (RM) | 0.035±0.022 | 0.026±0.024 | Ours (RM) | **0.796**±0.014 | **0.809**±0.010 | |
| Amazon Review | Avg Conf | 0.290±0.043 | 0.310±0.045 | MSP | 0.420±0.022 | 0.218±0.031 | |
| | Ens Avg Conf | 0.229±0.038 | 0.217±0.036 | Trust Score | 0.414±0.024 | 0.237±0.044 | |
| | Proxy Risk | 0.021±0.014 | 0.037±0.076 | Proxy Risk | 0.417±0.042 | 0.434±0.046 | |
| | Ours (RI) | 0.065±0.037 | 0.062±0.051 | Ours (RI) | 0.384±0.032 | **0.453**±0.037 | |
| | Ours (RM) | **0.018**±0.010 | **0.022**±0.011 | Ours (RM) | **0.426**±0.036 | 0.440±0.037 | |

Table 1: Results for unsupervised accuracy estimation and error detection. For typical DNN, We use CNN-BN for Digits, ResNet50 for Office-31, ResNet34 for CIFAR10-C, ResNet50 for iWildCam, and Fully Connected Network for Amazon Review. We show the mean and standard deviation of absolute estimation error and F1 score (mean±std). The numbers are calculated over the training-test dataset pairs in each dataset category. **Bold** numbers are the superior results.

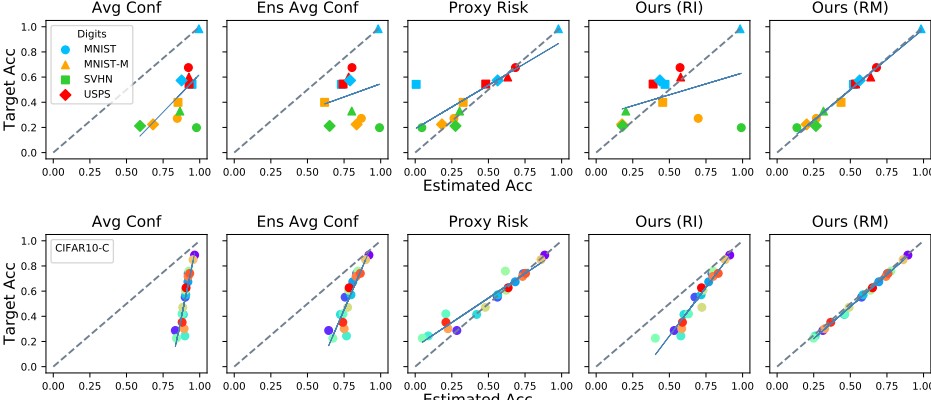

Figure 1: Accuracy estimation detailed results for each dataset pair from the dataset category Digits and CIFAR10-C (results for other categories are in the Appendix C.2). We use typical DNN as the architecture for the model $f$. We use symbols to represent training datasets and colors to represent test datasets. For CIFAR10-C, there is only one training dataset with multiple test datasets. The dashed line represents perfect prediction (target accuracy = estimated accuracy). Points beneath (above) the dashed line indicate overestimation (underestimation). The solid lines are regression lines of the results.

on Digits, by at least 25.6% on Office-31, by at least 2.0% on CIFAR10-C, by at least 4.7% on iWildCam and by at least 1.4% on Amazon Review. This shows the advantages of our method to identify error points in the unlabeled test dataset.

**Ablation Studies.** To study the effect of using ensembles, we vary the ensemble size $N$ ($N = 1$ means one single $h$, without ensemble) in our method with $\mathcal{T}_{RM}$. Similarly, for self-training, we vary the self-training iteration number $T$ ($T = 1$ means no self-training). Figure 2 shows their effect on the average F1 score: both ensemble and self-training techniques have positive effects for identifying error points and thus also for accuracy estimation. Moreover, increasing $T$ and $N$ can lead to further

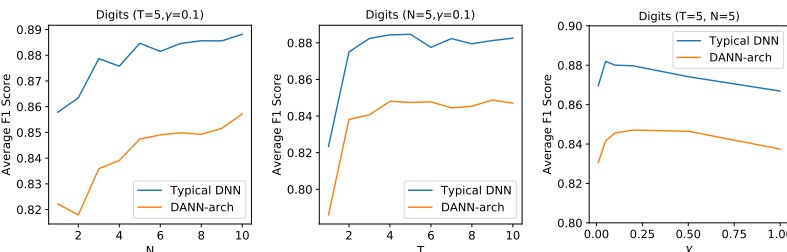

Figure 2: Ablation study for the effect of ensemble and self-training techniques on Digits (an additional ablation study on CIFAR10-C is in Appendix C.7). $N$ is the number of models in the ensemble, $T$ is the number of self-training iterations, and $\gamma$ is the weighting parameter for the loss term on the pseudo-labeled data. The ensemble training algorithm we use is $\mathcal{T}_{RM}$.

improvement. In addition to $N$ and $T$, we similarly exam the effect of the last hyper-parameter $\gamma$. The figure shows that a wide range of $\gamma$ can lead to good results, so it is easy to pick a suitable $\gamma$.

**Validating the Theoretical Analysis.** Our analysis relies on three conditions **(A)(B)(C)** stated in Section 5. Our experiments in Appendix C.4 show that empirically they are roughly satisfied. For example, for typical DNN $f$ on MNIST→MNIST-M, $\tilde{\nu} = 3.39\%$, $\tilde{\gamma} = 0.73\%$ while $\sigma_L^2 = 26.58\%$. We note that our theoretical analysis is for formalizing our intuition and is for the worst case. Even when our assumptions are not fully satisfied on some complex datasets, our method can still have empirical performance better than the error bound.

**Evaluating $f$ with Various Architectures.** We evaluate different architectures for the model $f$ on Digits and show the results in Appendix C.5. The results demonstrate that our method consistently outperforms other methods on pre-trained models with various deep learning model architectures.

**Analysis on Target Accuracy.** In Appendix C.8, we show that the target accuracy of our ensemble check models can be higher or lower than that of the model $f$ depending on the datasets and in both cases, our method can achieve good performance.

**Analysis on Proxy Risk.** In Appendix C.9, we show that the disagreement maximization is crucial for Proxy Risk and combining Proxy Risk with ensemble doesn't lead to better results than our method.

## 8   Discussions

While our framework is general and flexible when combined with different ensemble methods, and it is easy to design ensemble methods satisfying the success conditions, we note that some prior knowledge about the data is needed to achieve the best performance. For example, iWildCam has imbalanced classes which hurt representation matching and thus $\mathcal{T}_{RM}$, while $\mathcal{T}_{RI}$ is more suitable for such data. Different instantiations thus have their own limitations. For $\mathcal{T}_{RM}$, matching failure can cause errors on $f$'s correct points, and too strong matching can decrease the diversity since the models are too restricted. For $\mathcal{T}_{RI}$, it does not attempt to improve the accuracy on the test inputs and relies only on diversity to satisfy condition **(C)**. Then it can have worse performance than $\mathcal{T}_{RM}$ on test data with connection to the training data that can be exploited. The success conditions we identified can guide the design of different instantiations. In this paper we focus on ensembles of deep learning models, since they readily satisfy the conditions. However, other ensemble methods with other types of models may also be useful. We leave the exploration for future work.

## Acknowledgments

The authors would like to thank Dr. Ankur Taly for his valuable comments. The work is partially supported by Air Force Grant FA9550-18-1-0166, the National Science Foundation (NSF) Grants CCF-FMitF-1836978, IIS-2008559, SaTC-Frontiers-1804648, CCF-2046710 and CCF-1652140, and ARO grant number W911NF-17-1-0405. Jiefeng Chen and Somesh Jha are partially supported by the DARPA-GARD problem under agreement number 885000.

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
