# OpenReview forum: "Detecting Errors and Estimating Accuracy on Unlabeled Data with Self-training  Ensembles"
_NeurIPS.cc/2021/Conference — NeurIPS 2021 Poster_

### Official Review · Reviewer_rdjh · 2021-07-06

**Rating:** 7
**Confidence:** 3

**Summary:**

The paper studies the problem of error detection and accuracy estimation of unlabeled data points. Particularly, the paper proposes to use a diverse ensemble of models and detect errors by checking the inconsistency between the ensemble prediction and the target model prediction. Moreover, the paper proposes a self-training procedure that iteratively identifies incorrect predictions and adds their pseudo labels as new training targets to detect more errors. The paper also gives some theoretical analysis given assumptions about the ensemble models' quality.

**Ethical Concerns:**

No ethical concerns.

**Limitations And Societal Impact:**

They are properly addressed.

**Main Review:**

The proposed method matches my intuition about an ensemble of deep learning models. The empirical performance also looks very promising. The presentation of the paper is also very clear.

It may be worth investigating specific wrongly predicted data points and compare their true labels with their different predictions from f, models in the ensemble and across different self-training stages.

For the empirical verification about the theoretical analysis. I think the authors should also provide the final bound calculated using the estimated parameters.

Minor comments:
I don't quite agree to use "linear neural networks" to refer to the layers with fully connected network structures. I think it's more appropriate to call them fully connected layers as there are ReLU non-linearities. The linear neural network should refer to the networks without non-linearities.

## Post Rebuttal Update
Thanks to the authors for answering my questions in the rebuttal. After reading the rebuttal and discussing it with other reviewers, I decide not to change my rating of the paper. The major concern is that the approach is quite related to the Proxy Risk work. I suggest the authors give more analysis about the similarities and differences between the two works in the next version of the paper.

**Time Spent Reviewing:**

4

---

> ### Author Response · Authors · 2021-08-09
> **Thank you for the positive feedback!**
>
> We thank the reviewer for the valuable feedback, and would like to clarify the following points.
>
> **[Investigating specific wrongly predicted data points]**
>
> Thanks for the suggestion! We have partially done so for some datasets, in particular, iWildCam. The observations match our intuition there: on out-of-distribution test points, the pre-trained classifier makes errors while our ensemble can have large diversity on those points.
>
> **[Final estimated bound]**
>
> We note that our theoretical analysis is for formalizing our intuition, not for computing a numeric estimation of the performance. The error bound is for the worst case, and our method can have empirical performance better than the bound.
>
> **[Minor comments]**
>
> Thanks for the suggestion! We will change “linear neural network” to “fully connected network”.

---

> ### Author Response · Authors · 2021-09-14
> **Similarities and differences between our work and proxy risk work**
>
> Thanks for the update! We want to point out the similarities and differences between our work and proxy risk work, and we will include clarifications in the next version. Some of the following have already appeared in our responses to Reviewer 6eDi.
>
> **[Similarities]**
>
> 1. Both proxy risk and our framework train check models to estimate the accuracy of the pre-trained model f on the unlabeled test dataset U and also identify misclassified points in U;
>
> 2. Both proxy risk and one instance of our framework (Algorithm 3) use domain-invariant representations (DIR) to improve the accuracy of the check models on the target domain.
>
> **[Differences]**
>
> 1. The key ideas are different. The idea of ProxyRisk is to find a check model with maximum disagreement in the good hypothesis class (the set of hypotheses that achieve small DIR loss). Our idea is to **increase** the disagreement on the **misclassified points** in each iteration of self-training, and misclassified data points are identified by either accurate prediction or diversity using ensemble.
>
> 2. The training objectives are different.
>
>     (1) ProxyRisk's objective is applied on the whole unlabeled test set U. Ours is only on the selected subset of U (i.e., the currently identified misclassified points).
>
>     (2) The terms in the objective to encourage disagreement are different. Proxy risk tries to maximize the disagreement between the model f and the check model h directly on the entire unlabeled test set U while maintaining a small DIR loss (corresponding to the term $-\mathbb{E}\_{x \in U\_X}\ell(f(x),h'(x))$ in their objective). In contrast, our method encourages disagreement via fitting the check models to the pseudo-labeled dataset R (corresponding to the term $\mathbb{E}\_{(x,y)\in R}\ell(h(x),y)$ in our objective).
>
>     The two terms $\mathbb{E}\_{(x,y)\in R}\ell(h(x),y)$ and $-\mathbb{E}\_{x \in U\_X}\ell(f(x), h'(x))$ are different. For multi-class classification, disagreeing with the pre-trained model is not equivalent to agreeing with the pseudo-labels. This is because the check model can predict some labels different from both the pre-trained model's prediction $f(x)$ and the pseudo-label $\tilde{y}_x$. For example, suppose there are 3 classes, and suppose the pre-trained model predicts $f(x)$= class 0,  the pseudo-label $\tilde{y}_x$=class 1. Our term is ensuring $h(x)$ to be class 1, while their term is ensuring $h'(x)$ to be either class 1 or class 2. As has been pointed out in our previous reply, our objective is more specific, and this leads to increasing disagreement and also potentially better prediction from the ensemble, which then leads to the success of the self-training.
>
> 3. The implementation of the objectives are also different. The proxy risk method uses $L_2$ norm $-\mathbb{E}\_{x \sim U\_X} [\ell(f(x), h'(x))]=\mathbb{E}\_{x \sim U\_X} [ - || \bar{h'}(x) - \bar{f}(x) ||\_{2} ]$ (see their code on Github: [proxy_risk.py#L52](https://github.com/chingyaoc/estimating-generalization/blob/master/proxy_risk.py#L52)). While our method uses cross entropy loss $\mathbb{E}\_{(x,y)\in R}[\ell(h(x),y)]=\mathbb{E}\_{(x,y)\in R} [-\log \bar{h}(x)_{y}]$.

---

### Official Review · Reviewer_1YgW · 2021-07-11

**Rating:** 7
**Confidence:** 3

**Summary:**


The paper proposes a framework to tackle two tasks: 1. unsupervised accuracy estimation (estimate accuracy on unlabeled data) and 2. error detection (identify the misclassified examples in the unlabeled data). The framework uses an ensemble of networks to identify mis-classified points as points where the current classifier disagrees with the ensemble. Then the mis classified points are used for self-training to improve the ensemble. The paper gives a theoretical analysis which depends on assuming that the ensemble is mostly correct where the classifer is correct, the classifier and the ensemble disagree on pseudolabeled data and that the ensemble is diverse in its predictions, which is claimed to be true empirically.

**Limitations And Societal Impact:**

The authors note that some prior knowledge is needed to achieve the best performance, and this will influence the design of the ensemble method, which remains an art. Social impact is not discussed.


**Main Review:**

Quality

The framework is simple and makes good sense, the experiments are reasonably extensive, and there is some theoretical understanding, although it depends on strong conditions on the ensemble. As long as the ensemble obeys good properties on possibly OOD data, the framework seems to make sense, but there seems to be an art to this. For example, the conceptual understanding doesn't address the distribution shift between training and unlabeled examples, which is perhaps why Algorithm 3 (RM) incorporates a domain adaptation method and in general surpasses Algorithm 2 (random init ensemble) by a large margin consistently.


- Self-training seems to improve the performance a lot, as in Figure 2. If self-training is also combined with Proxy Risk, does the ensembling method still improve over it on the 5 datasets? This would make clearer the role of ensembling vs self-training.
- Although the theoretical assumptions are somewhat strong, the paper does show empirically that the values in the conditions are favorable on the digit datasets. What about in the more complex datasets?
- In general, the ablations of ensembling and self-training parameters are only on Digits - do the same trends hold for other datasets?
- When do the conditions hold? For example, what if the pre-trained classifier itself is a majority vote over an ensemble?
- There are some observations that aren't reported with numbers, such as in L147.


Significance

Unsupervised error detection and estimation are interesting tasks that, if we can make progress on them, it would help in model development for domain adaptation/OOD generalization, as well as a score for model selection, and as a tool in algorithmic frameworks for improving the model such as the one presented in this paper.


Originality
The main novel idea is to use ensembles + self-training in a similar framework to the Proxy Risk paper. The use of domain invariant representations, a check model, and disagreement is in similar spirit to Proxy Risk.
- Neural network ensembles have been used for uncertainty quantification / OOD detection, but the error detection task here is slightly different as it takes into account the performance of some classifier, and the goal is to detect examples with errors.
- The ensemble method in Algorithm 3 is reminicient of other domain adaptation + semi-supervised learning methods, for example DIRT-T.

Clarity

The paper is mostly clearly written.
- the definitions of the various sets R_X / W_X / B_X / G_X etc are scattered throughout and hard to remember.
- In Figure 1, the color/shapes change from the legend. I think this is denoting new train/unlabeled pairs, but this is pretty subtle.

**Time Spent Reviewing:**

4

---

> ### Author Response · Authors · 2021-08-09
> **Thank you for the positive feedback!**
>
> We thank the reviewer for the valuable feedback, and would like to clarify the following points.
>
>
> **[Distribution shift]**
>
> The distribution shift is implicitly captured by our conditions.
>
> When there is a distribution shift that still allows accurate prediction (e.g., by domain adaptation), a properly constructed ensemble can have good predictions to satisfy our conditions. This is discussed in Line 123 and captured by our conditions (A)(C). A concrete example is MNIST$\to$MNIST-M.
>
> When there is a distribution shift with significant out-of-distribution (OOD) test examples that cannot be classified correctly, the ensemble can have large diversity on those OOD points. This is discussed in Line 127 and captured by our condition (C). A concrete example is iWildCam.
>
> **[Self-training vs ensembling]**
>
> To separate their effects, we can consider self-training without ensembling and ensembling without self-training. These are actually shown in Figure 2 (when N=1 or T=1).  The results show that both techniques have positive effects.
>
> It is unclear how to combine self-training with proxy risk since they’re designed based on different considerations. Our self-training aims to get disagreement on the misclassified points, not directly to increase the target accuracy. On the other hand, proxy risk first pre-trains an accurate check model h and then fine-tunes it to maximize the disagreement. One can use self-training to pre-train h but that does not align with the design of proxy risk.
>
> **[Verifying assumptions on other datasets]**
>
> We further verify our success conditions on the Office-31 dataset category:
>
> 1. on amazon$\to$dslr, $\tilde{\nu}=6.16$%, $\tilde{\gamma}=1.92$%, $\sigma^2_L=15.61$% and the estimation error is 0.0181;
>
> 2. on dslr$\to$webcam, $\tilde{\nu}=2.17$%, $\tilde{\gamma}=0.20$%, $\sigma^2_L=8.80$% and the estimation error is 0.0088.
>
> We further note that our theoretical analysis is for formalizing our intuition and is for the worst case. Even when our assumptions are not fully satisfied on some complex datasets, our method can still have empirical performance better than the error bound.
>
> **[Ablation study on other datasets]**
>
> We perform ablation study experiments on the CIFAR10-C dataset category and observe a similar trend. We will add more ablation study results to the paper.
>
> **[When the pre-trained classifier itself is a majority vote over an ensemble]**
>
> We perform an experiment on MNIST$\to$MNIST-M to study the case where the pre-trained classifier is a majority vote over an ensemble. The ensemble contains 10 models trained with different random initializations. The results are similar to those for a single model. Our success conditions are still roughly satisfied and the accuracy estimation is great: $\tilde{\nu}=2.61$%, $\tilde{\gamma}=0.59$%, $\sigma^2_L=28.94$% and the estimation error is 0.0009.
>
> **[Empirical evidence to support observations]**
>
> We report the numbers in Appendix B.4 to support the observation stated in L147.
>
> **[Originality]**
>
> We would like to point out that our framework is different from existing methods like proxy risk.
>
> 1. The principle of our framework is to get disagreement on misclassified points, achieve the disagreement by either accurate prediction or diversity, and improve it by self-training. While that of proxy risk is to ensure each model is accurate in the check model class and maximize the disagreement over the class.
>
> 2. Our use of ensembling+self-training is different from the standard ones. In particular, our goal is to get disagreement, not directly improve accuracy (though this can happen). While the standard use aims at accuracy.
>
> Please refer to the first item in our response to Reviewer 6eDi for details.

---

> > ### Comment · Reviewer_1YgW · 2021-08-14
> > **response**
> >
> > Thanks for the clarifications, which answered many of my concerns, and I raise my score to 7. Looking at some of the other reviews, I think that the final paper would be greatly improved by including some of the discussion on 1) differences with proxy risk and perhaps ablations to differentiate the components. The fact that we mostly look for disagreement instead of accuracy is interesting. 2) more extensive experiments (like in the review) on verifying the theoretical conditions, and 3) more deeply examining what actually happens in the pseudolabels / ensemble predictions in the datasets to confirm if the behaviors are as expected. Some of these are addressed in parts of the author rebuttals throughout.

---

### Official Review · Reviewer_6eDi · 2021-07-14

**Rating:** 5
**Confidence:** 3

**Summary:**

The paper considers the problem of unsupervised accuracy estimation and error detection of pre-trained classifiers. The contributions consist of (i) a bound on the generalization performance on estimated accuracy and on detected errors and (ii) a solution based on ensemble models. Specifically, the bound on the generalization error takes into account three quantities, namely (i) the error of the ensemble on correctly-classified test instances, (ii) the agreement of the ensemble and the pre-trained model on potentially wrong test instances and (iii) the diversity of the ensemble on mis-classified test instances. The solution based on ensemble models consists of an iterative training strategy, where at each iteration the ensemble is trained to minimise a training surrogate for quantity (i) combined with quantity (ii) and ensuring diversity among ensemble predictions across iterations. Experimental analysis on several benchmarks (including digits, Office-31, CIFAR-10, iWildCam and Amazon Reviews) show the potential of the proposed solution.

Overall, the paper is well written and well structured. The idea of using ensemble strategies in the context of unsupervised accuracy estimation and error detection is novel. However, there are some issues in terms of ORIGINALITY and QUALITY, which need to be addressed and/or discussed (see below for further comments).

**Limitations And Societal Impact:**

There is no paragraph/section addressing the societal impact. Is there a specific reason for that? As far as I understand, the considered problem has a security impact, as being fundamental for the deployment of deep learning models to real-world scenarios.

**Main Review:**

ORIGINALITY:
The idea of using ensemble methods is interesting and novel in the context of unsupervised accuracy estimation and error detection. However, the work is incremental (extending [1] to the ensemble setting) and it is not clear up to which extent we can consider it novel. Can the authors highlight the main algorithmic and theoretical differences with respect to [1], as there is no discussion in the main article?

QUALITY:
Overall, the paper looks sound (I haven't checked the proof though).
Regarding the theoretical bound (Theorem 1), the assumptions on the three quantities, namely small error rate, low agreement and high diversity over all training iterations of the ensemble, are quite strong. How can these conditions be guaranteed in practice and for contexts where the test domain has small relation with the training one? Further experimental analysis on benchmarks other than digits (currently shown in the Supplementary) would provide a better support to the theory.

Regarding the experiments, most of the claims about the superiority of the proposed strategy over competitors are overly and improperly stated. In fact, note that in Table 1, the work of [1] has high variance in almost all cases. Are the performance of your solution statistically significant compared to the ones of [1]?
Furthermore, how are the hyperparameters of the proposed solution (i.e. T, gamma, alpha, N) chosen?

CLARITY:
The paper is well written and well structured.

SIGNIFICANCE:
The problem is significant and relevant for future deployment of deep learning models to real-world cases. However, the significance of the proposed solution is low, as it is not clear up to which extent the theoretical findings and the experimental observations can provide additional knowledge to the understanding of the problem.

[1] Chuang et al. Estimating Generalization under Distribution Shifts via Domain-Invariant Representations. ICML 2020


**Time Spent Reviewing:**

6

---

> ### Author Response · Authors · 2021-08-09
> **Clarifying the novelty and contributions**
>
> We thank the reviewer for the valuable feedback, and would like to clarify our contributions.
>
>
> **[Novelty over proxy risk]**
>
> Our framework is built based on different principles from proxy risk, and the use of  ensembling+self-training is different from the standard ensembling+self-training.
>
> 1. We first briefly review the proxy risk method.
>
> It has two stages: first build a highly accurate check model with domain adaptation, and then maximize the disagreement by fine-tuning the check model. Both stages are motivated by their Lemma 4: the difference between the proxy risk and the true risk is bounded by the max risk of the check model class on the test inputs, where the proxy risk is the maximum disagreement between the check model and the pre-trained classifier. Therefore, the principle is aiming to get a check model class with small risks (=high accuracy), and get the maximum disagreement within the class, leading to the two stages. See the discussion in Section 4 of their paper.
>
> 2. We now clarify our principle.
>
> As discussed in the intuition in Section 4, we aim to get disagreement on misclassified points. We achieve this disagreement in two ways: accurate prediction or diversity.
>
> (1) Getting disagreement on misclassified points is different from maximizing disagreement within the check model class. We aim at misclassified points, and we don’t try to maximize within the model class.
>
> (2) Either accurate prediction or diversity can serve our goal of disagreement on misclassified points. In contrast, proxy risk aims to get high accuracy for all models in the check model class.
>
> (3) We further use self-training to improve the disagreement and thus identify more and more misclassified points.
>
> Both ensembling and self-training are novel for the tasks of accuracy estimation and error detection. Furthermore, ensembling and self-training are not simply adopting standard ones to the tasks, as detailed below.
>
> Note that since we can use either accurate prediction or diversity, our framework doesn’t need to be restricted to domain adaptation as in proxy risk. Rather, domain adaptation is only used in one concrete instantiation of the framework.
>
> 3. Our ensembling and self-training is different from standard ensembling and self-training.
>
> (1) The goals are different.
> Standard self-training+ensemble aims to get accurate predictions. Ours is to increase disagreement on misclassified points by either getting accurate predictions or getting diversity (Line 120-148). More precisely, as discussed in Line 128, ensembling allows diversity on points that we cannot have good predictions; and in Line 137, self-training makes sure we have more and more disagreement and thus identify more and more misclassified points, where the disagreement can be obtained by either correctness or diversity of the ensemble. This is formalized in Condition (C): either correct or diverse on W_X \ R_X.
>
> (2) The techniques are also different, due to the different goals.
> (a) For each identified misclassified point, we assign a pseudo-label that is different from the prediction of the pre-trained model, but we do **not** need the pseudo-labels to be correct (see Line 138-142), as our goal is disagreement. In contrast, standard methods typically hope the pseudo-labels are correct.
> (b) We only assign pseudo-labels to the currently identified set of misclassified points, while standard methods typically assign pseudo-labels for all unlabeled points. This is because we want disagreement on misclassified points rather than on all points, and we would like to assign pseudo-labels to be different from the prediction of the pre-trained model on misclassified points.
>
> 4. We have also provided theoretical analysis to formalize our intuition and improve our understanding of the tasks.
>
> Section 5 shows that our above idea can lead to provable guarantees. Condition (A) is to justify why we aim to get disagreement on misclassified points (as discussed in Line 112-121),  Condition (C) corresponds to the two ways of either accurate prediction or diversity (as discussed in Line 122-135), and Condition (B) is for the success of self-training (as discussed in Line 143). We also provide additional discussion in Appendix A.2 on the benefit of using ensembles and on the connections to proxy risk and calibration. Roughly speaking, we show that using ensembles allows more general and tighter bounds than proxy risk and calibration.
>
> 5. In summary, our algorithmic contributions are:
>
> (1) ensemble for detecting errors and estimating accuracy;
>
> (2) self-training for detecting errors and estimating accuracy;
>
> (3) a general framework with flexible instantiations.
>
> Our theoretical contributions are:
>
> (1) theoretical justification for our algorithmic contributions;
>
> (2) enhance the understanding of the tasks by pointing out either accuracy or diversity can help error estimation using disagreement;
>
> (3) enhance the understanding by pointing out self-training with iteratively improving the estimation.
>
>
> **[Significance of experimental results]**
>
> 1. We want to point out that the variance in Table 1 is not about the variance of the results under different runs with different random seeds, but the variance of the performance across different training-test dataset pairs in each dataset category. The high variance of the results of the proxy risk method means the performance of the proxy risk method is not stable across the training-test dataset pairs. In our experiments, we observe that sometimes the proxy risk method will fail miserably.
>
> 2. We want to highlight that the performance of our method is statistically significant compared to the proxy risk method. We have provided both mean and standard deviation in Table 1, which show that our advantage is statistically significant. We have also provided detailed results in Figure 1, which also clearly shows our advantage across dataset pairs. Further results are provided in Figure 3 and 4 in the Appendix.
>
> **[Assumptions]**
>
> We note that our success conditions are not strong and can be satisfied in practice. We further verify our success conditions on the Office-31 dataset category: on amazon$\to$dslr, $\tilde{\nu}=6.16$%, $\tilde{\gamma}=1.92$%, $\sigma^2_L=15.61$% and the estimation error is 0.0181; on dslr$\to$webcam, $\tilde{\nu}=2.17$%, $\tilde{\gamma}=0.20$%, $\sigma^2_L=8.80$% and the estimation error is 0.0088. On the other hand, we note that our theoretical analysis is for formalizing our intuition, not for computing a numeric estimation of the performance. The error bound is for the worst case, and our method can have empirical performance better than the bound.
>
> In comparison, the success condition of proxy risk is that all models in the check model class have small risks on the test data, which is harder to be satisfied (formalized in our discussion in Appendix A.2). Their bound is also for the worst case.
>
> Finally, for the case when the test domain has a small relation with the training one, we cannot get high accuracy but our condition can still be satisfied via diversity. For example, iWildCam test data have out-of-distribution data, on which we don’t have accurate predictions but have diversity, thus satisfying our Condition (C). In contrast, proxy risk fails to get high accuracy and thus gets worse performance.
>
> **[Choosing hyper-paramters]**
>
> The hyper-parameters $T$, $N$, $\gamma$, can be easily selected since a broad range of values can lead to good results. Based on our observation, larger $T$ and $N$ can lead to better results. In our paper, we simply select $T=5$ and $N=5$ by considering the computational cost. For the hyper-parameter $\alpha$, we follow the previous work (proxy risk paper) to set it.

---

> > ### Comment · Reviewer_6eDi · 2021-08-28
> > **Concerns**
> >
> > Dear Authors,
> >
> > thank you for your reply. I went through the ProxyRisk paper and I still have some concerns about its similarity with your work:
> >
> > 1. the connection between ProxyRisk and your work seems direct. Indeed, line 4 in Algorithm 1 is equivalent to maximise the disagreement between the check and the pre-trained model. Note also that the training of the ensemble is equivalent to the objective of ProxyRisk (compare the line after the for loop in Algorithm 1 of ProxyRisk paper and eq. 4 in this paper). The overall objectives are indeed the same. The only remarkable difference is the use of an ensemble and self-training and I acknowledge this difference. Is that correct? Otherwise, can you point out what are the differences in terms of objective function?
> > 2. I still feel that the experiments are not fully conclusive and fair. In particular, would the result be the same by comparing against ProxyRisk and an ensemble model? This would help to clarify what is the real benefit of your strategy.

---

> > > ### Author Response · Authors · 2021-08-28
> > > **Clarifying misunderstandings**
> > >
> > > Thanks for the response! We believe there are some misunderstandings and we are happy to clarify them below.
> > >
> > > **[Q1]**
> > >
> > > First, line 4 in our Algorithm 1 is not about maximizing the disagreement. The idea behind our algorithm is to **increase** the disagreement on the **misclassified points** in each iteration of self-training. We select those points that we are confident to be misclassified and add them to the pseudo-labeled set. Then when we train the next ensemble, we make sure the next ensemble fits the pseudo-labels, but do not maximize disagreement (this is explained in detail below when discussing our eq. (4) in Algorithm 3). ProxyRisk doesn't select points and does directly maximize the disagreement.
> > >
> > > Second, our objective to train the ensemble using $\mathcal{T}\_{RM}$ (eq. (4) in Algorithm 3) is different from the maximizing disagreement objective of ProxyRisk (the line after the for loop in Algorithm 1 of ProxyRisk paper). Although $R\_S(f’g’)+\alpha \cdot d(p\_S^{g’}(Z), p\_T^{g’}(Z))$ in the ProxyRisk’s objective is the same as $\mathbb{E}\_{(x,y)\in D}[\ell(h(x),y)]+\alpha \cdot d(p\_D^{\phi}, p\_{U\_X}^{\phi})$ in our objective, $-R\_{T}(h,h’)$ in the ProxyRisk’s objective is very different from $\mathbb{E}\_{(x,y)\in R}[\ell(h(x),y)]$ in our objective. Under our notations, $-R\_{T}(h,h’)$ is equivalent to $-\mathbb{E}\_{x \sim U\_X} [\ell(f(x), h’(x))]$. So ProxyRisk tries to make the check model $h’$ disagree with the pre-trained model $f$ on every data point in $U\_X$ while maintaining a small DIR loss $\mathbb{E}\_{(x,y)\in D}[\ell(h’(x),y)]+\alpha \cdot d(p\_D^{\phi’}, p\_{U\_X}^{\phi’})$. Compared to ProxyRisk, we use a different way to encourage disagreement: we make the model $h$ fit the pseudo-labeled set $R$ ($R\_X$ is a subset of $U\_X$). Thus, the objectives are quite different.
> > >
> > > **[Q2]**
> > >
> > > We believe our experiments are conclusive and fair. Our understanding of your question is that what if ProxyRisk is applied with a check model that is an ensemble. However, it is unclear how to apply ProxyRisk on an ensemble check model. ProxyRisk's disagreement maximization step is not defined for an ensemble check model. In fact, the goal of ProxyRisk is to find **a check model** in the good hypothesis class (the set of hypotheses that achieve small DIR loss) that achieves maximum disagreement. Thus, ProxyRisk may not be compatible with an ensemble check model.

---

> > > > ### Comment · Reviewer_6eDi · 2021-08-30
> > > > **Concerns are Still Valid**
> > > >
> > > > Thank you for the clarifications. Please, find here some considerations.
> > > >
> > > > **Comparison of objectives between ProxyRisk and yours**
> > > >
> > > > Indeed, the two objectives are the same, except for the computation of the disagreement term. In ProxyRisk the disagreement is computed on the whole unlabelled set, while for your objective the disagreement is computed on a subset of the unlabelled data.
> > > > This difference is due to the fact that you are introducing the self-training strategy and I acknowledge this difference. However, the concern about the similarity of the objective still remains.
> > > >
> > > > **Ensemble**
> > > >
> > > > I do not agree that "it is unclear how to apply an ensemble to ProxyRisk". Indeed, you can always define an ensemble model even for ProxyRisk. For example, you can consider $h'(x)=\text{Ensemble}(h_1'(x),\dots,h_{t-1}'(x),h_{t}'(x))$ and train with respect only to the current model $h_{t}'(x)$. This is similar to what is done in your Algorithm 3.
> > > >
> > > > Considering these aspects, I would like to see a comparison against ProxyRisk with an ensemble. Without that, there is no enough evidence to claim the difference between ProxyRisk and your method and more importantly to state the superiority over ProxyRisk.
> > > > Therefore, I keep my score.

---

> > > ### Author Response · Authors · 2021-08-31
> > > **Clarifying misunderstandings**
> > >
> > > Thanks for the quick reply! We believe there are still some misunderstandings as explained below.
> > >
> > > **[Objectives]**
> > >
> > > The term $\mathbb{E}\_{(x,y)\in R}[\ell(h(x),y)]$ is different from the disagreement term $-\mathbb{E}\_{x \sim U\_X} [\ell(f(x), h’(x))]$ in ProxyRisk.
> > >
> > > In the implementation, $\mathbb{E}\_{(x,y)\in R}[\ell(h(x),y)]=\mathbb{E}\_{(x,y)\in R} [-\log \bar{h}(x)_{y}]$ while $-\mathbb{E}\_{x \sim U\_X} [\ell(f(x), h’(x))]=\mathbb{E}\_{x \sim U\_X} [ - || \bar{h'}(x) - \bar{f}(x) ||\_{2} ]$. Here, $\bar{h}$ is the softmax output of $h$ (similarly for $\bar{h'}$ and $\bar{f}$). So in the implementation, the two terms are very different.
> > >
> > > The key differences are:
> > >
> > > 1. As agreed by the reviewer, ours is computed on a carefully selected subset instead of on the whole unlabeled set.
> > >
> > > 2. More importantly, $\mathbb{E}_{(x,y)\in R}[\ell(h(x),y)]$ is about agreement with pseudo-labels, which is different from disagreement between the pre-trained model $f$ and the check model $h’$. The two may look similar since pseudo-labels $\tilde{y}_x$ are different from the prediction of the pre-trained model $f(x)$. But the two actually have a key difference for multi-class classification: disagreeing with the pre-trained model doesn’t mean agreeing with the pseudo-labels, as the check model can predict some labels different from both $f(x)$ and $\tilde{y}_x$. In other words, our objective is more specific, trying to ensure $h(x) = \tilde{y}_x$. As has been explained in our intuition, this leads to increasing disagreement and also potentially better prediction from the ensemble, which then leads to the success of the self-training.
> > >
> > > **[Ensemble]**
> > >
> > > The suggested method gets worse performance than our method.
> > >
> > > We have implemented the suggested method: the check model $h’(x)$ is an ensemble of models $h’\_1(x),\dots, h’\_t(x)$ via majority vote and each model $h’\_j(x)$ in the ensemble is trained using ProxyRisk’s maximizing disagreement training objective (similar to what is done in our Algorithm 3). We set $t=5$ and use typical DNN as the model architecture of $f$. The experimental result on the Digits dataset category is: the mean absolute estimation error is 0.0872 and the average F1 score is 0.8309. The performance of the suggested method is worse than that of our method (mean estimation error: 0.0230, average F1 score: 0.8810).
> > >
> > > The experimental results have again shown the effectiveness of our method. On the other hand, we would like to point out that it’s unclear if the suggested method matches the key idea of ProxyRisk. The subtlety is that maximizing the disagreement of each model in this greedy way is not the same as maximizing the disagreement of the whole ensemble.

---

> > > > ### Comment · Reviewer_6eDi · 2021-08-31
> > > > **Concerns are Still Valid**
> > > >
> > > > Thanks for the further explanation. However, my doubts are still there.
> > > >
> > > > Indeed:
> > > > 1. The two terms $E_{(x,y)\in R}\ell(h(x),y)$ and $-E_{(x,y)\in U_x}\ell(f(x),h'(x))$ are the same (except for the fact that you are using two different subsets to compute the expectation). The first term is minimising the agreement between the prediction $h(x)$ and the pseudo labels, which is equivalent to **maximize the disagreement** between the ensemble and $f(x)$ (as the pseudo labels correspond to the majority vote of the ensemble, which is different from $f(x)$.
> > > > 2. You are using a different loss for your case and ProxyRisk, using cross-entropy in the first case, while using L2 loss in the second one. It is widely known that **training a DNN using a L2 loss** does not perform well as when using a cross-entropy. Therefore, I cannot trust the experimental results provided in this last answer.
> > > > 3. Note that in the proposed modification of ProxyRisk using the ensemble, the **label always correponds to the majority vote** of the ensemble. Let us recall the definition $\text{Ensemble}[h_1'(x),\dots,h_{t-1}'(x), h_t'(x)]$. If the majority vote of $\text{Ensemble}[h_1(x)',\dots,h_{t-1}'(x), h_t'(x)]$ is equal to $f(x)$, then $h_t(x)$ will be modified to ensure that the output of the ensemble is different from $f(x)$ and the both the output of the ensemble ("the pseudo-labels") and $h_t(x)$ are the same. In the opposite case, namely $\text{Ensemble}[h_1'(x),\dots,h_{t-1}'(x), h_t'(x)]$ is different from $f(x)$, $h_t'(x)$ does not affect the final result and therefore also the loss. Consequently, the loss, when useful, will always drive the current predictor to adhere with the majority vote of the ensemble (namely "to adhere with the pseudo-labels").

---

> > > ### Author Response · Authors · 2021-08-31
> > > **Clarifying misunderstandings**
> > >
> > > Thanks for the quick reply! However, there are still some misunderstandings.
> > >
> > > **[Objectives]**
> > >
> > > The two terms $\mathbb{E}\_{(x,y)\in R}\ell(h(x),y)$ and $-\mathbb{E}\_{x \in U\_X}\ell(f(x),h’(x))$ are different, even considering that the pseudo-label is different from $f(x)$, and ignoring the difference in subsets and L2/cross-entropy losses. Please check the subtle details below.
> > >
> > > For multi-class classification, disagreeing with the pre-trained model **is not equivalent to** agreeing with the pseudo-labels. This is because the check model can predict some labels different from both the pre-trained model’s prediction $f(x)$ and the pseudo-label $\tilde{y}_x$. For example, suppose there are 3 classes, and suppose the pre-trained model predicts $f(x)$= class 0,  the pseudo-label $\tilde{y}_x$=class 1. Our term is ensuring $h(x)$ to be class 1, while their term is ensuring $h’(x)$ to be either class 1 or class 2. As has been pointed out in our previous reply, our objective is more specific, and this leads to increasing disagreement and also potentially better prediction from the ensemble, which then leads to the success of the self-training.
> > >
> > > **[L2 loss]**
> > >
> > > **L2 loss is the loss used by the implementation of ProxyRisk.** Please check their code on Github: [proxy_risk.py#Line 52](https://github.com/chingyaoc/estimating-generalization/blob/master/proxy_risk.py#L52).
> > >
> > > As we are making comparisons with ProxyRisk, we believe we should follow what the original ProxyRisk is using. Whether training a DNN using the L2 loss is worse or not is not something related to our discussion here.
> > >
> > > Anyway, we are also curious about why ProxyRisk used L2 loss, and have run experiments of ProxyRisk with cross-entropy instead of L2 loss. It turned out that ProxyRisk using the L2 loss gets better performance than using cross-entropy loss. The experimental results on the Digits dataset category when using typical DNN as the model architecture of $f$ are: when using L2 loss, the mean absolute estimation error is 0.0850 and the average F1 score is 0.8440, while when using cross-entropy loss, the mean absolute estimation error is 0.3452 ​and the average F1 score is 0.6464.
> > >
> > > **[Ensemble]**
> > >
> > > This matches what we meant: we only sequentially update the models in the ensemble (i.e., don’t update $h\_1$ to $h\_{t-1}$ when updating $h\_t$). The method is greedy search, which is different from finding the ensemble that maximizes the disagreement. So the suggested method deviates from the key idea of ProxyRisk.

---

### Official Review · Reviewer_o4cm · 2021-07-21

**Rating:** 6
**Confidence:** 3

**Summary:**

- The paper considers two problems: unsupervised accuracy estimation and error detection, both in out-of-distribution settings.
- The authors propose a practical approach, where they train a ‘check model’ h using self-training and ensembling. Points on which the prediction model and the check model disagree on are predicted as misclassified.
- The theoretical analysis identifies conditions under which the approach should work and presents a bound on the accuracy estimation error
- The authors empirically show that their algorithm outperforms baselines for both unsupervised accuracy estimation and error detection

**Limitations And Societal Impact:**

Yes

**Main Review:**

- Overall, the proposed approach is well-motivated, supported by theoretical analysis, and has thorough experiments. My main comments are about potential areas of improvements in the experiments.
--------------

- Empirically, the proposed approach outperforms many existing baselines on multiple datasets. Beyond the benchmark numbers, the paper also includes thorough experiments with additional empirical analysis, ablations, multiple versions of the algorithm, etc. But I think there are potentially a few caveats with the experiments as described below.
- The paper has an ablation experiment, where they study the effect of ensembling by changing the number of models in the ensemble. One potential confounding factor is that the check model may be more accurate with more models in the ensemble, and this may be particularly relevant since better-performing check models are expected to work better. Is ensembling important, even when controlled for check model accuracy/size?
- The paper compares the proposed approach with multiple baselines, but it seems like many of the baselines do not use unlabeled test examples, whereas the proposed approach does (through self-training). Is this correct? Do you think it would be more appropriate to compare with other methods that use unlabeled test examples? For example, https://arxiv.org/abs/2012.05825 also considers detection of misclassified points in OOD/transductive setting, using an ensembling technique.
Another ablation I’d be interested in is on the specific self-training procedure proposed. In contrast to the standard self-training methods, the proposed approach pseudolabels only a specific subset of the dataset. How does this compare with standard pseudolabeling + ensembling?
- The paper says the approach works when the ensemble check model h is either accurate or diverse on the points misclassified by f. It also mentions that self-training can sometimes improve the accuracy of h, or increase the diversity of the predictions on the points misclassified by f but not yet identified by h. To what extent are the empirical gains explained by the high accuracy of h, as opposed to the diversity in the ensemble?
- The last few bullet points would give me more confidence about the novelty of the paper. My understanding is that the proposed algorithm is a nice combination/modification of well-known techniques, and the work considers a relatively unexplored setting, outperforms existing baselines, and has theoretical analysis that’s new to my knowledge. However, I think the contributions would be more incremental if it turns out that the high accuracy of h is the main reason why the approach works, and the high accuracy of h is achieved by standard self-training and ensembling; training a check model has been proposed before, and self-training and ensembling are well-known ways to improve model performance.
- In general, how do the prediction accuracies of f and h compare?
--------------

- The paper is clearly written, and I enjoyed reading it.
- Additional discussion on how the three assumptions may or may not hold up in practice could be helpful. The paper describes them as minor assumptions that’s likely to hold in practice, but it’s not clear to me why/if that’s the case for most datasets/models.
- It seems like there is a difference between the analyzed framework and the practical approach, and I wasn’t sure why this was the case. Why does algorithm 1 use majority voting to select R (lines 4 and 5) as opposed to following the procedure in Framework 1 (line 4)?
----------------------
Minor comments
- N seems overloaded between Algorithms 1 and 2
- I found the discussion of DANN-arch confusing. What’s the difference between typical DNN and DANN-arch without the discriminator head, and what’s the rationale? Is the important part the fact that h is trained with DANN in one, but not the other, rather than the architectural differences?
- How did you select the hyperparameters? The paper describes the chosen values (e.g., “we set T=5,...”), but it’s unclear how they were selected.
- Line 301: Figure 3 → Figure 1


**Time Spent Reviewing:**

3

---

> ### Author Response · Authors · 2021-08-09
> **Clarifying our contributions**
>
> We thank the reviewer for the valuable feedback and would like to clarify the following points.
>
> **[High accuracy vs diversity]**
>
> In our intuition (Line 122-148) and theoretical analysis, we point out that disagreement on misclassified points is the main reason for our strong performance, and this disagreement is achieved by two ways: either accurate prediction or diversity.
>
> 1. We emphasize that diversity is a crucial element for our advantage.
> Empirically, there are experiments where the accuracy on the test inputs is not high but our error estimation performance is great. For example, on SVHN->USPS, the target accuracy of the ensemble model with majority vote after self-training is only 67.46%, but the estimation error is 0.1%. This strong empirical performance follows from diversity but not high accuracy. Theoretically, besides the analysis in Section 5, we provide additional discussions in Section A.2 on using diverse ensembles showing its advantage over a single check model and calibration.
>
> 2. We also emphasize that our method is not simply improving the accuracy of existing check model methods like proxy risk.
> Consider again the above example of SVHN->USPS. In comparison, the pre-trained check model used by the proxy risk method has a higher target accuracy of 69.86% than ours, but their estimation error is 4.83% which is much worse.
> Besides, we want to point out that the high accuracy of the check model h is also not the only reason why the proxy risk method works. Proxy risk has two stages: first train a check model and then fine-tune it to maximize the disagreement. Since it is usually hard to train a check model h with very high target accuracy, disagreement maximization also plays an important role. For example, without disagreement maximization, the mean F1 score of proxy risk on Digits will decrease from 0.844 to 0.812. Compared to proxy risk, our method uses a novel way to encourage disagreement and get better performance (this contribution is orthogonal to the proxy-risk method).
>
> 3. We also would like to point out that our ensembling and self-training aims at improving disagreement but not on accuracy, which is different from the standard ensembling and self-training. This is detailed in our next bullet item.
>
> 4. To what extent are the empirical gains by the high accuracy or the diversity depends on the dataset.
> On datasets where one cannot obtain high accuracy (e.g., with out-of-distribution data that cannot be correctly classified like iWildCam), diversity is essential for good performance. On other datasets, we do not see a consistent trend: the target accuracy of our method can be lower or higher than the proxy-risk method, while our performance is typically better than the proxy-risk method.
>
>
> **[Differences from standard self-training and ensemble]**
>
> Our self-training ensembles method is different from the standard self-training and ensemble.
>
> 1. The goals are different.
> Standard self-training+ensemble aims to get accurate predictions. Ours is to increase disagreement on misclassified points by either getting accurate predictions or getting diversity (Line 120-148). More precisely, as discussed in Line 128, ensembling allows diversity on points that we cannot have good predictions; and in Line 137, self-training makes sure we have enhanced disagreement and thus identify more misclassified points, where the disagreement can be obtained by either correctness or diversity of the ensemble. This is formalized in Condition (C): either correct or diverse on W_X \ R_X.
>
> 2. The techniques are also different, due to the different goals.
> (1) For each identified misclassified point, we assign a pseudo-label that is different from the prediction of the pre-trained model, but we do **not** need the pseudo-labels to be correct (see Line 138-142), as our goal is disagreement. In contrast, standard methods typically hope the pseudo-labels are correct.
> (2) The reviewer has already noticed that we only assign pseudo-labels to the currently identified set of misclassified points, while standard methods typically assign pseudo-labels for all unlabeled points. This is because we want disagreement on misclassified points rather than on all points, and we would like to assign pseudo-labels to be different from the prediction of the pre-trained model on misclassified points.
>
>
> **[Novelty]**
>
> We believe that our contributions are novel.
> As discussed above, our method relies on disagreement from both accuracy and diversity, instead of simply improving the accuracy over existing methods. Furthermore, our method achieves disagreement by novel self-training and ensembling, which is designed based on our own consideration rather than simply using the standard self-training and ensembling. All these come along with theoretical justification and empirical support.
>
>
> **[Assumptions]**
>
> We note that our success conditions are not strong and can be satisfied in practice. We further verify our success conditions on the Office-31 dataset category: on amazon$\to$dslr, $\tilde{\nu}=6.16$%, $\tilde{\gamma}=1.92$%, $\sigma^2_L=15.61$% and the estimation error is 0.0181; on dslr$\to$webcam, $\tilde{\nu}=2.17$%, $\tilde{\gamma}=0.20$%, $\sigma^2_L=8.80$% and the estimation error is 0.0088.
>
> The intuition why Condition (A) is satisfied is described in Line 176. Condition (B) is satisfied when the ensemble can roughly fit the pseudo-labels, which is easily true for powerful models. Condition (C) depends on if the ensemble method can have diversity on out-of-distribution points, which is typically true for deep network ensembles (the reason for this is still an active research question.)
>
> We further note that our theoretical analysis is for formalizing our intuition and is for the worst case. Even when our assumptions are not fully satisfied on some complex datasets, our method can still have empirical performance better than the error bound.
>
>
> **[Other baselines]**
>
> 1. We want to point out that the proxy risk method also uses unlabeled test examples.
> 2. The paper by Alexandru et al (https://arxiv.org/pdf/2012.05825.pdf) is for detecting out-of-distribution inputs, not for detecting misclassified inputs. For example, they consider all examples in the CIFAR10-C as out-of-distribution data for the in-distribution dataset CIFAR-10, while we would like to identify points in CIFAR10-C that get misclassified.
>
> **[The accuracy of f and h]**
>
> On some datasets where the domain adaptation method can work well (e.g. Digits), the accuracy of the check models h built by $\mathcal{T}_{RM}$ on test inputs is usually higher than that of the pre-trained model f. In such cases, the high accuracy of h plays an important role in the good performance. However, on some datasets where the domain adaptation method fails (e.g. iWildCam), the accuracy of h is typically not higher than that of f. In such cases, our method can still achieve good performance due to the diversity of the ensemble.
>
> **[Choosing hyper-paramters]**
>
> The hyper-parameters $T$, $N$, $\gamma$, can be easily selected since a broad range of values can lead to good results. Based on our observation, larger $T$ and $N$ can lead to better results. In our paper, we simply select $T=5$ and $N=5$ by considering the computational cost. For the hyper-parameter $\alpha$, we follow the previous work (proxy risk paper) to set it.
>
> **[The way to select R]**
>
> We have explained the reason why we select R via majority vote briefly in lines 233-234: these two methods lead to similar performance since the data points that cause a difference are rare (detailed justification is provided in Appendix B.6).
>
> **[DANN-arch]**
>
> DANN-arch means the model f shares the same architecture as the check model h. But the training methods for training f and h may be different. In the proxy risk paper, they also consider such a case where f and h share the same architecture. Thus, we include experimental results for this to make a better comparison.

---

> > ### Comment · Reviewer_o4cm · 2021-08-23
> > **Response**
> >
> > Thank you for the clarifications and the thorough response. It addressed many of my concerns, and I've updated the score.

---

### Decision · Program_Chairs · 2021-09-27

**Decision:**

Accept (Poster)

**Comment:**

The paper proposed a novel approach that leverages ensembles and self-training for unsupervised accuracy estimation and error detection. All reviewers find the problem setup interesting and the paper well written, with theoretical justification (although relying on strong assumptions) and reasonable empirical support. There are some useful suggestions during the discussion phase, in particular for improving the experimental results.

After a few rounds of interaction during the discussion phase, the committee reached a consensus on the technical contributions: Reviewers agree upon the technical novelty as using ensembles and self-training for improving model disagreement -- a subtle but different approach to the existing work of ProxyRisk, which warrants the novelty of this work; it is worth mentioning that reviewers also note that the use of domain invariant representations, a check model, and disagreement are in a similar spirit to prior work.

An initial disagreement among the committee was on the scope of the experiments: whether a modified version of ProxyRisk with an ensemble is necessary to be included. Although the modification suggested in the discussion phase (by Reviewer 6eDi) deviates from the ProxyRisk algorithm proposed in the original work, the committee agree that this could be viewed as an additional ablation of the role of ensembling vs self-training, which would have made the work stronger. The authors are encouraged to take into consideration of such feedback when preparing a revision.